# Strategies for Monitoring Microbial Life in Beach Sand for Protection of Public Health

**DOI:** 10.3390/ijerph20095710

**Published:** 2023-05-03

**Authors:** João Brandão, Elisabete Valério, Chelsea Weiskerger, Cristina Veríssimo, Konstantina Sarioglou, Monika Novak Babič, Helena M. Solo-Gabriele, Raquel Sabino, Maria Teresa Rebelo

**Affiliations:** 1Department of Environmental Health, National Institute of Health Dr. Ricardo, Avenida Padre Cruz, 1649-016 Lisboa, Portugal; elisabete.valerio@insa.min-saude.pt (E.V.);; 2Centre for Environmental and Marine Studies (CESAM), Department of Animal Biology, University of Lisboa, Campo Grande 016, 1749-016 Lisboa, Portugal; 3Department of Civil and Environmental Engineering, Michigan State University, 1449 Engineering Research Ct. Room A127, East Lansing, MI 48824, USA; 4Department of Transmittable Diseases, National Institute of Health Dr. Ricardo, Avenida Padre Cruz, 1649-016 Lisboa, Portugal; 5Department of Biology, Biotechnical Faculty, University of Ljubljana, Jamnikarjeva 101, 1000 Ljubljana, Slovenia; 6Department of Chemical, Environmental, and Materials Engineering, University of Miami, 1251 Memorial Drive, Coral Gables, FL 33146, USA

**Keywords:** sand, recreational water, fungi, FIB, MST, monitoring program

## Abstract

The 2021 revised guidelines of the World Health Organization recommend monitoring the quality of sand in addition to water at recreational beaches. This review provides background information about the types of beaches, the characteristics of sand, and the microbiological parameters that should be measured. Analytical approaches are described for quantifying fungi and fecal indicator bacteria from beach sand. The review addresses strategies to assess beach sand quality, monitoring approaches, sand remediation, and the proposed way forward for beach sand monitoring programs. In the proposed way forward, recommendations are provided for acceptable levels of fungi given their distribution in the environment. Additional recommendations include evaluating FIB distributions at beaches globally to assess acceptable ranges of FIB levels, similar to those proposed for fungi.

## 1. Introduction

A day at the beach is spent mainly in the sand area where people lie down sunbathing and where children play. The wind displaces loose grains of sand, some of which end up deposited on beachgoers skin and hair or forced into the ear, nose, and mouth. At the end of that day, beachgoers return home, inevitably taking some sand and microorganisms along with them. Yet, despite the higher amount of time spent on sand than in water, the latter has well established acceptable levels of microbes based upon exposure, risk, and safety standards [1], while sand is lagging in such characterizations [2,3].

The rationale that we spend more time on the sand than in the water was clearly recognised by the World Health Organization (WHO) in 2003, within Chapter 6 of the Guidelines for Safe Recreational Water Environments [3]. Between 1969 and 2003, many publications addressed microbes in beach sand. Yet, it was not until 2012 that the health impacts of sand exposure were quantified. In the 2012 publication, Heaney et al. [4] analyzed 144 wet sand samples for the presence of the fecal indicator bacteria (FIB) enterococci and conducted 4999 interviews describing contact with sand. The study found that enterococci in sand was associated with gastrointestinal (GI) illness. Beachgoers who dug in the sand or were buried in sand exhibited higher incidence rates of GI compared with those who did not.

There are other organisms, such as *Cryptosporidium* spp., *Clostridium perfringens*, and *Bacteroides* spp., which have been used to indicate fecal contamination of bathing waters. The latter are used mainly to identify the microbiological source of fecal pollution [5,6]. Fujioka and collaborators [7], discussed the recreational water quality criteria of 2012 of the United States of America (USA), and concluded that quality indicators would need to be adjusted in many regions to reflect emerging pathogens. For example, the regulations of the State of Hawai’i included use of *C. perfringens* as a FIB for decades. The rationale behind the use of *C. perfringens* as a fecal indicator in tropical waters is based upon the fact that *C. perfringens* requires anaerobic conditions to multiply outside of the human host and is therefore unable to multiply in aerobic surface soils and waters. The more mainstream FIB, such as fecal coliform, *E. coli*, and enterococci, multiply in the environment under warmer and more humid environmental conditions. Considering climate change, alternative indicators such as *C. perfringens* may become more mainstream to control for environmental regrowth as temperatures increase [2,7,8].

One of the current approaches recommended in 2004 by the WHO for drinking water is the establishment of water safety plans (WSPs), to minimize threats to water supplies by systematically assessing and managing risks [9]. This recommendation covers all possible scenarios where humans can be exposed to microbes, including sediment. The European Commission readily adopted this perspective for the most recent Bathing Water Directive 2006/7/EC [10]. In the WHO 2020 review of the guidelines, WSPs were also recommended for recreational waters [11]. This emphasizes the need to consider the nearshore environment as a possible source of microbial exposure, inclusive of pathogens and opportunistic organisms.

## 2. Sand Characteristics and Surrounding Environment

### 2.1. Types of Sand and Artificial Beaches

Th beach is defined as the zone of unconsolidated material between the low water line and the landward limit of wave swash, often marked by either a change in landform or vegetation [12]. Exposed sandy beaches are physically dynamic habitats, inhabited by specialized biotic assemblages that are structured mainly by physical forces [13]. Beach sand is composed of minerals and water that can sustain entire micro-ecosystems that comprise a wide variety of biological forms. Microorganisms embed within complex biofilms that are attached to the surfaces of the grains of sand, thriving on the available nutrients [2,14]. In 2016, Abreu and collaborators published a paper that compared sand grain size and composition with the microbial community [15]. This research group found that sand granulometry and chemical composition was not significantly associated with microbial concentration. However, Valério et al. [16] verified that different amounts of sand are needed for deoxyribonucleic acid (DNA) extraction, depending on the grain size, to achieve equivalent yields.

Furthermore, Abreu et al. [15] compared natural and artificial beaches, revealing a difference in microbial concentrations. Manmade structures, built to maintain sand in place, impeded the natural wave activity and thus limited wash-off of microorganisms from sand, resulting in microbial accumulation in the supratidal area. Coincidentally, this is the area mainly used by beachgoers for sunbathing and relaxing. Similarly, Hernandez et al. [17] found that sand mineralogy may be related to its ability to retain microbes, with quartz sand and smoother surfaces retaining fewer microbes relative to carbonate sands with rougher surfaces and higher surface areas.

Beach nourishment is a common practice for reclaiming coastlines lost to wave activity and natural erosion and for building artificial beaches [17,18]. The microbial community of the sand used for renourishment should be considered since it will carry its own native microbial communities to the new location [19]. When the artificial beach *Praia da Calheta* was built and nourished with sand originating in the Sahara, Morocco, the first batch of sand brought live scorpions along with it. Consequently, the project had to be halted and redesigned [20]. Nourishing sand should thus originate as geographically close as possible to its final location, in order minimize the introduction of non-native species. In addition, to minimize losses by erosion and to maximize the chances of success of nourishment projects, the sand used to nourish beaches is recommended to have grain-sizes 1.5–2.0 times those of the original sand [18].

### 2.2. Swash Zone vs. Supratidal Zone

The sand area of a beach is divided into several sections (Figure 1). In this text, the focus shall be on the swash zone (the intertidal area limited above by the high tide reach) and on the supratidal zone (beyond wave and high tide reach). The latter tends to stay dry unless impacted by precipitation and run-off from the backshore. Backshore run-off is a well-known cause of sand contamination, as demonstrated by an episode of a destructive tropical storm hitting the island of Madeira in 2010 [15]. The storm destroyed facilities of all kinds, including the sanitary infrastructure. Due to the heavy rain, land- and mudslides rushed down to the lower lands along the coastline. FIB from this surge could be detected within the beach sand for approximately three months after the storm event [21].

Conversely, the swash zone is influenced by the waves and tides [2]. A review focused on the micropsammon (bacteria, fungi, parasites, and viruses) fate and transport describes how sand microbial communities include microbial species that are both indigenous and allochthonous (i.e., non-native) to the local environment [22]. The review also addressed the naturalisation of allochthonous micropsammon, introduced by water, air [23], run-off, seaweed [24], and by animals. For reference, FIB are considered allochthonous to beach sand [25].

## 3. Microbial Parameters

Prediction and mitigation of contaminant surges at beaches requires an understanding of microbial communities in sand, their inter-species interactions in multi-communal biofilms, and their use of molecular warfare against other susceptible species, with antibiotics and statins, for example [2,26,27,28]. The first approach to maintaining a balanced level of microbial life in sand is to prevent contamination in the first place by implementing hygiene measures, e.g., regular trash removal, providing restroom facilities for beachgoers, removing remains of fish from the sand, etc. However, these actions may not always be possible. Brandão et al. [29] and WHO [11] discuss possible actions to take for remediation in case of necessity (included in Table 1).

Biofilms provide an interesting additional dynamic to controlling microbial communities in sand. In the laboratory, the biofilms may reduce the colony forming units extracted from sand due to agglomeration. However, FIB in biofilms would not be readily available as inocula to form new colonies in the natural environment. To visualize, Brandão et al. [28], presents pictures of initial stages of biofilm construction on the surface of sand grains, as revealed by electron microscopy. Results of the visualization show that biofilms are complex, multi-species communities that can integrate fungi with other microorganisms. For example, when black mold is a constituent of a biofilm exposed to sun radiation, its presence will protect other species from the damaging effects of ultra-violet (UV) radiation by absorbing it as a source of energy for its own growth. As a carbon source for its growth, black mold will use either decaying biofilm inhabitants or hydrocarbons present from fossil-fuel residue or shedding by more complex lifeforms visiting or inhabiting the beach [31].

### 3.1. History of Microbial Tracking in the Sand

The first paper published on beach sand contaminants was in 1960 by Schönfeld, Rieth and Thianprasit, and it described a study which aimed to detect the presence of dermatophytes in a Baltic Sea resort. The study yielded only the geophilic dermatophyte *Arthroderma insingulare* (formerly *Trichophyton terrestre*) in the superficial layers of supratidal sand [32]. In 1973, however, Gertrud Müller in a wider study confirmed the absence of anthropophilic dermatophytes in sands. For this research, Müller’s team sampled Estoril, near Lisboa, the Adriatic Sea, at Gabicce Mare and at Grömitz, on the German Baltic coast. The team searched for *Epidermophyton floccosum*, an anthropophilic species, for three years, and sampled twice per week (in the same places). They also aimed to find beach usage-associated variations of this fungal species [33]. The study revealed a clear surge in the presence of *Epidermophyton floccosum* in sands touched by human bare feet. It failed to isolate other species, common in other kinds of soil, showers, and pools such as *Trichophyton interdigitale* (formerly *T. mentagrophytes var. interdigitale*) or for *Nannizzia gypsea* (formerly *Microsporum gypseum*). The authors pointed out in their report that the inability to isolate these common agents of dermatophytosis should not rule out their presence in beach sand that had been in contact with human feet.

Maria Colon Valiente, in 1990, hypothesized that the absence of dermatophytes in beach sand was likely due to the low nutrient environment or the high maximum temperatures [34]. These observations, however, were challenged by Sousa [35], who reported dermatophytes in 11 out of 24 (45.83%) beaches studied in coastal areas around Lisboa. Brandão et al. [36] sampled all regional coasts of Continental Portugal every 2 months, for 13 months, and generated 210 samples of sand from both the supratidal and the vadose zones. The three types were wild beaches, beaches with water quality problems and beaches awarded blue flag status for exceptional quality. Despite their high maintenance levels in relation to the other beach types, blue flag beaches did not exhibit the lowest fungal concentrations in sand. In this study, a higher density of beach users correlated with higher levels of dermatophytes during the summer months. Not only was the presence and concentration of fungi in beach sand highly variable, but the fungal community may have been also evolving due to land use and climate changes. Therefore, additional research and monitoring will be integral to characterizing and maintaining beach sand quality in the context of fungal contaminants. Recently, *E. floccosum* seems to be decreasing in its prevalence in clinical specimens worldwide, indicating that the likelihood of isolating it from beach sand may be diminished [37]. Other dermatophytes, and even other microbial life forms, may alter their patterns of existence due to clinical intervention and climatic alterations [2].

Oshiro and Fujioka [38] studied the bacteriological water quality of Hanauma Bay in Hawai’i. Due to the extreme human densities associated with recreational use, water quality of Hanauma Bay was difficult to maintain at acceptable levels. The Bay has since been classified a nature preserve and is open to recreation during limited timeframes, to allow the local ecosystem to recover without the threat of constant high human densities [39]. Because of its status as a bay full of marine wildlife, and frequent use by tourists and local beachgoers, the water quality led Oshiro and Fujioka to sample the shoreline water and sand, land runoff, and mongoose and pigeon feces, to try to find the main causes of the surges of FIB. The samples were analyzed for fecal coliforms, *Escherichia coli*, and enterococci, and revealed that the major sources of the periodically high levels of these bacteria in the water were contaminants founds in beach sand, namely pigeon feces [38]. In this case, water quality was the main driver of the research, although the source came from the sand.

In 2017, Argentina became the first nation to formally include sand inspection for rubbish in its water quality standards [40]. Lithuania added monitoring of helminths in sand to their 2007 National regulation within the 2018 updates [41]. The Lithuanian regulations are currently the only known regulation on helminths, as mentioned in WHO [11].

During the last 20 years, several publications have identified potential microbial culture parameters for sand monitoring [3,42,43,44]. Others contemplate meta-genomics, raising possible alternative parameters, based on nucleic acid analyses [21,45,46,47]. However, until there is a regulatory document defining sand quality parameters, the choice for monitoring parameters is likely to fall on a combination of culturable fecal indicator organisms (FIO), to indicate pathogens and opportunists associated with wastewaters, and others that do not relate to fecal pollution (including most fungi). Next generation sequencing (NGS) may be an interesting screening tool to assess variations in the microbiota as shown in Taylor and Kurtz [47]. In this study, the authors evaluated three beaches along the Grand Strand of South Carolina, USA. The authors sequenced the V4 region of the 16S rRNA gene to compute relationships between diversity and temporal or local factors. Gammaproteobacteria, Planctomycetes, Acidobacteria, and Actinobacteria were the dominating bacterial populations at these beaches. The communities were similar in overall composition and diversity, but the abundance of taxa changed over time.

Table 2 shows pathogens, opportunistic microorganisms, and fecal indicators in beach sand currently available in the literature.

The following three sections describe the current taxa of interest for sand monitoring. WHO [11] addresses three other biological groups of interest that are currently poorly represented in the scientific literature, namely viruses, helminths, and insects.

### 3.2. Fecal Indicator Bacteria (FIB)

Fecal pollution remains one of the biggest problems with recreational waters and beach sand. Parts of the globe have taken action to eliminate the discharge of untreated sewage into water bodies. However, in other regions, fecal contamination of bathing and even drinking water sources remains a serious problem. One of the most notorious contamination episodes of water took place in 2010 in Haiti, due to the destruction caused by an earthquake of magnitude 7.0. This event killed an estimated 230,000 people and injured another 300,000. Haiti already had a relatively low coverage of sanitation infrastructure but after the earthquake, only 10% of the rural population and 24% of the urban population had access to improved sanitation [57]. This low rate of access to drinking water and sanitation coupled with contamination of the water supply led to an outbreak of cholera, which resulted in 658,563 reported cases and 8111 deaths from the disease as of 2 June 2013 [58].

The WHO guidelines for safe recreational water environments [3] recommends the use of FIB to detect fecal pollution in recreational waters. The more recent guidelines [11], extend those recommendations to beach sand, precisely to avoid waterborne diseases originating from human excreta. The recommendations focus on measurements of FIB, not the actual pathogens. According to the literature, exposure to 40 CFU/mL of enterococci results in an illness probability of 1%, and up to 200 CFU/100 mL results in up to 5% probability of illness. This indicator parameter in recreational waters suggests general disease probability rather than any specific pathogen-associated illness. It also is currently in use in regulations, e.g., the European Bathing Water Directive [10]. Fujioka et al. [7], Weiskerger et al. [2], and Teixeira et al. [8] discuss the need to extend beyond FIB in the future, to accommodate changes arising from climatic alterations. Looking into pathogens directly instead of quantifying indicators is not only increasingly easy, but will also have to be implemented for pathogens that in the future may not be predicted through the use of FIB.

### 3.3. Other Bacteria

As described in WHO [3], Sabino et al. [43], and Weiskerger et al. [2], controlling exposure to bacteria to protect human health needs to go beyond FIB, since the supratidal zone of a beach may not be greatly contaminated with fecal pollution. Instead, its main source of pollution is often skin shedding from animals, vegetable debris, and other organic matter that may serve as food for wildlife [36].

The following bacterial taxa have been considered of relevance for the protection of human health: *Staphylococcus* spp. (skin infections), *Vibrio* spp. (cholera and necrotizing fasciitis), *Clostridium perfringens* (food poisoning, possible cause of bacteraemia), *Campylobacter jejuni* (gastroenteritis), *Shigella* spp. (haemorrhagic diarrhoea), and *Pseudomonas aeruginosa* (superficial and systemic infections) [3,43,59]. Methicillin-resistant *Staphylococcus aureus* (MRSA) was reported in 2009 as present in beach sand and water in California [60,61] and Florida, USA [62] and more recently in South Africa, in 2015–2016 [63], presenting yet another emerging contaminant of concern for beachgoers. This group of sand contaminants and climate change implications is addressed further by the WHO [11] and in Weiskerger et al. [2].

### 3.4. Fungi

Most fungi are opportunistic, which means that exposure is a potential risk only for susceptible individuals. Yet, two groups of fungi are of great concern: dermatophytes and endemic fungi. Dermatophytes are keratinophilic fungi that cause dermatophytosis, superficial infections of the hair, nails, skin, and scalp. There are anthropophilic, geophilic, and zoophilic dermatophytes and they are so classified according to their transmission route: if transmission is human-to-humans, soil-to-human, or animal-to-human [64]. The current knowledge suggests that anthropophilic dermatophytes may be susceptible to the environmental conditions in beach sand and thus may die rather quickly [65]. This does not necessarily mean that there is no transmission at the beach, though. Shedding takes place naturally, especially from infected areas of the skin that become dry and scaly, and death due to radiation or high temperatures is not immediate. In fact, under laboratory conditions, recreational beach sands from Hawai’I were able to maintain several fungal species. In the study, Anderson [66] tested the survival of *Cutaneotrichosporon cutaneum/Trichosporon asahii, Candida albicans, Nannizzia gypsea* and *Trichophyton mentagrophytes* (var. *mentagrophytes)* and all survived for at least one month in non-sterile sand inoculated with keratinized propagules. These data suggest that non-detection of anthropophilic dermatophytes may be associated more with the natural dispersion of propagules and representativeness of sampling, rather than the dermatophyte survival itself.

As for the endemic fungi in sensu stricto (*Coccidioides* spp., *Paracoccidioides brasiliensis*, *Histoplasma* spp., *Blastomyces dermatitidis*), no known publications indicate their isolation from beach sand. However, their natural habitats suggest some degree of survival in beaches within endemic areas (inland, mainly). *Coccidioides* spp. inhabits dry, desert-like territories of the USA, and of Argentina. The remaining fungi are endemic to more humid habitats such as the Mississippi River Valley, USA (*Blastomyces* spp.); central and eastern USA (*Histoplasma capsulatum*); Sub-Saharan Africa (*Histoplasma duboisii);* and South America (*Paracoccidioides brasiliensis*) [67]. *Cryptococcus deuterogattii* is also endemic to the Pacific Northwest of the North American continent [68] but not considered dimorphic (real pathogens). There is yet to be consensus whether this species should be considered simply a genetic type of *Cryptococcus gatii* (type AFLP6/VGIIa).

Other fungi of interest, as listed recently by WHO [11] for exposure in natural environments, are *Mucorales*, allergenic fungi, and dematiaceous fungi. *Mucorales* is the order that includes the fungi responsible for the invasive mucormycosis. This order has raised some concern due to the opportunistic ability to start an infection in immunocompetent individuals when inoculated deeply under the skin by piercing materials. This has been well demonstrated by a cluster of cases of necrotizing cutaneous mucormycosis following a tornado in Joplin, Missouri, USA in 2011 [69] where several individuals were injured by deep puncture wounds. *Mucorales* are also a relevant group of fungi in cases of near drowning, as described by Sympardi et al. [70].

However, the distribution of fungal species has been reportedly different in different contexts. Cogliati et al. [71] found different distributions of yeasts and molds in beaches throughout Europe depending on the latitude and heavy metal composition of soil. Yeasts seem to withstand low temperatures better during winter than molds and are therefore more prevalent in Northern European coasts. Moreover, molds are more associated with soil rich in nickel and yeasts with soils rich in cadmium. This results in a distribution of fungi mainly at the deltas of European rivers and lagoons, where these metals accumulate in river sediment.

Considering that an allergy is host-dependent and that there are individual allergies to most fungi, for the purpose of this text, all fungi are considered to also cause allergies. There are, however, different types of allergies: respiratory and contact allergies. Respiratory allergies imply inhaling allergens and therefore are mainly caused by airborne-sporulating fungi. The extent to which fungal spores can travel airborne has been clearly described by Kellogg and Griffin [72], in an open report about the air travel of fungi from Africa to North America. Fungal spores can be airborne and can also extend their presence to an entire beach. Although information on fungal inhalation leading to infections specifically from sand is currently unavailable, Buskirk et al. [73] showed in a murine model that the dry exposure to 10^5^ spores of *A. fumigatus* twice per week triggers an inflammatory response in the lungs within 24 and 48 h. Additionally, an IgG elevation is observed after seven days, concomitant with spore germination. Tanaka et al. [74] found that cytokine release in immunocompetent individuals takes place about 19 h post-inhalation. Cho et al. [75] observed in fungal respiratory deposition models that particles of *Stachybotrys chartarum* might be deposited in numbers 230–250-fold higher than spores. These data suggest a delayed first response to the exposure to fungal allergens, but an existing one, nonetheless. Beach users might thus not even associate an allergic episode with a visit to the beach the day before. Lastly, exposure to volatile organic compounds (VOCs) produced by fungi like 3-Methylfuran may cause nonspecific symptoms, namely eye, nose and throat irritations, headaches, and fatigue [76]. Aleksic et al. [77] described the difference between the propagation of aerosolized mycotoxins (mycophenolic acid, sterigmatocystin, and macrocyclic trichothecenes) of three common fungal indoor contaminants originating in the environment: *Penicillium brevicompactum, Aspergillus versicolor*, and *Stachybotrys chartarum*. Considering the outdoor dispersion of any biochemical molecules, the toxicogenic traits of fungi should not represent a relevant threat at the beach.

Dematiaceous or melanized fungi are taxa that produce melanin to harness energy from radiation to use in biochemical paths. The most common ailments associated with melanized fungi are keratitis as well as cutaneous, subcutaneous, and respiratory tract infections. Exposure to sand combined with a traumatic event may result in an invasive fungal infection (phaeohyphomycosis). The severity of the infection, though, depends on the extension of the trauma and immune response of the host [78,79].

Most recently, antifungal resistant *Candida auris*, an emerging thermo- and halotolerant, multi resistant yeast, has been recognized as an environmental concern in beach environments [49,80,81].

## 4. Sampling Time and Sampling Methods

Recreational beaches are typically maintained for health and safety only during official bathing seasons. Sampling outside the bathing season serves academic purposes and can eventually help develop beach management plans to improve beach quality during the bathing season, e.g., cleaning the beach, planning the number of trash-bins, beach nourishment to increase dispersion of sunbathers, mitigation of climate change effects, and removal of animal excrements. Sampling both sand and water seems like the most reasonable plan to manage the beach, allowing a full view of simultaneous snapshots to inter-relate and assess any directionality of distinct pollution in sand and in water together. Regulations should thus recommend sampling both sand and water and leave the decision of sampling outside the bathing season to the beach managers.

Water dynamics ensure that pollutants will experience high rates of diffusion facilitating the homogeneity. Sampling water at one site is thus often representative of a large volume. Sand, conversely, is patchy and thus sampling and representativeness should be considered for developing effective monitoring procedures [29,82,83]. The sampling frequency and the way samples are collected and transported to a laboratory for processing should be defined with a standard approach that would render monitoring efficient and equivalent between different communities with regard to the comparability of results obtained. Brandão [29] and WHO [11] have addressed this subject, with a sand monitoring recommendation.

Brandão et al. [36] described how the swash zone connects water with sand of the intertidal area of a beach. The authors conclude that the microbes from the intertidal sands are represented by nearshore water, which is monitored by default, due to regulation. So, sand only needs to be monitored in the supratidal area. Brewer et al. [84] demonstrated that single grab samples are representative of only the sample under analysis in the lab. That is the reason why an incremental approach (Incremental Sampling Methodology [85]) for soil sampling, although complex, is highly recommended. In this method, a decision unit that represents a significant area of volume or soil, is sampled in a regular grid pattern. Typically, at least 30 equal-mass samples are collected from the decision unit and replicates can be collected to assist in statistical analysis [85]. For locations with no historical information to help with decision making, Brandão [29] recommends a combination of incremental sampling and previous knowledge of hotspots of microbial contamination. This would allow for sampling of worst-case scenarios and thus render the sampling of sand relevant and representative of a beach. Should incremental sampling not be possible, a composite can be collected for beaches less than a few hundred meters in length. The composite sample would consist of three grab samples spanning the length of the beach. These three grab samples can then be combined, homogenized, and analyzed as one sample [50,85]. If beaches are more than a few hundred meter in length, the beach should be divided into sub-areas and each sub-area considered a separate beach.

## 5. Analytical Approaches for Characterizing Microbial Communities of Beach Sand

Analytical methods are issued by the International Standards Organization (ISO), upon adoption or voted in from of a pool of possible candidate methods, and after a robust screening of all possible scenarios. The European Commission for example, issues Water Quality Directives listing a reference method and proven equivalent methods. Each individual member-state may opt for another unlisted method, if it passes a three-expert panel that contemplates reproducibility, equivalence with the reference method and representativeness. Upon acceptance, the study must be replicated in other member-states that wish to integrate that non-reference method in their own list of options [86]. Different regions of the globe will always present specific features in public health protection from exposure to microbial life because they must address locally endemic pathogens. In the tropics, for example, the existence of percutaneous infection by nematodes, e.g., hookworm and schistosomiasis, can be a serious problem for beach users [87,88].

### 5.1. Two Tiers Analytical Approach

Solo-Gabriele et al. [19] recommended a two-tiered approach to sand quality monitoring, aiming to simplify the analytical approaches. Some of the methods described in the literature were extremely laborious and costly, which does not work well for routine analysis and to respond with beach management actions. Thus, the authors recommend a fast routine analysis, able to be performed in water quality laboratories and to escalate to reference laboratories only when necessary to establish the source of an outbreak or to investigate the source of a highly prevalent contaminant.

#### 5.1.1. Analysis during Outbreak Conditions

In case of an outbreak, namely the one reported by Brandão et al. [28], the Reference Mycology Laboratory was engaged to identify the relevant species of fungi. For this study, the fungal species isolated were typically associated with either vegetable matter (colonizers and pathogens) or with faucal contamination, which was indeed the cause of the outbreak. The local team described how a tropical winter storm delivered enormous amounts of near coastal debris into the cove, including high quantities of decaying vegetable matter. The debris was reflected in the fungal analysis, by the considerable presence of *Fusarium* spp., of *Aspergillus* section *Circundati*, and some *Candida tropicalis*, which are common plant pathogens and colonizers.

Other teams have considered analyzing the total microbiome using NGS [21,46,47]. This approach requires sophisticated analytical procedures and equipment, and the results are not comparable with culture-based methods. Despite analyzing the DNA present and thus detecting both viable and non-viable individuals, it is a good tool for environmental forensic analyses, since it provides microbial community composition information. One drawback is that the most successful species will be over-represented and may conceal the less represented taxa in NGS assays.

Information on routine analytical approaches for viruses (mainly molecular, but also some culture based), protozoa, and helminths is extremely scarce, so they shall not be addressed further.

#### 5.1.2. Quality Assessment Schemes

Once the sample of sand arrives at the laboratory, analysis is perform to ensure reproducibility and repeatability regardless of the equipment used. The efficiency of the aqueous extraction of the sand, plating of the extract, plate reading, and registering with the necessary calculations to produce a value of CFU/g are a few examples. Boehm et al. [89] published an article on sand bacterial analysis and performance of participating laboratories in an interlaboratory collaborative study. The authors found that there is variation in extraction efficiency between blending, shaking, and by analyst. No analysts were rejected from the study. Instead, they contributed to establish the natural variation of the results. Rinsing, decanting, and settling before using the eluent did not show any statistically significant variation (*p* < 0.05).

Participating in an interlaboratory assessment scheme is thus extremely relevant for sand analysis. It is the only tool that can ensure that the results obtained are independent of the analysts while serving as a training tool. Many laboratory analysts work alone and only know how consistently they perform. Distributing samples for interlaboratory assessment schemes implies processing, transporting, and handling of samples in transport to the end-user laboratory. The logistics involved have the potential to alter the sample in many ways. Non-refrigerated transport may either kill or permit multiplication of microorganisms and processing and handling might contaminate samples. Analysis of the joint results will be informative of all these possible sources of variability. The international standard used for determining a consensus value in proficiency testing by interlaboratory comparison is the ISO 13528 [86].

#### 5.1.3. Outbreak Response

In the absence of historical data on culturable microorganisms from beach sand, a primordial full-population-exploratory analysis may be an attractive approach. It would shortcut years of collecting scattered fragments of data to eventually generate a beach profile and be able to decide what is ordinary and what may require attention or management [29]. However, without a history and baseline knowledge of microbial sources at a site, a relative surge of any organism is impossible to confirm. Alternatively, parametric reference values may be used as guidance for never-before tested sands, but these are yet to be published by any regulating agency. WHO [11] provided the first regulatory document with sand microbial parameters and reference values for enterococci and fungi. It chose a total fungal count value from the work of Brandão et al. [65] to be indicative for beach management purposes rather than a parameter value. The historical data of a beach can reveal the results associated with influencing events (e.g., windstorms, heavy rain events and beach festival parties), as described in Brandão [29] and Brandão et al. [28].

In the case of a beach associated outbreak by any microbial agent, an epidemiological study, such as the one described in Brandão et al. [28], should be conducted. In such an event, the authors recommend the analysis of many types of variables, one of them being if the beach sand might be the cause of the outbreak. To evaluate the cause of the outbreak, bathing was immediately excluded as a contributor to the outbreak because some of the infected patients did not bathe. The only common denominator to the reported macular erythematous pruritic rash outbreak was sand. As the cause was unknown during the investigation, the study was conducted, employing organic chemistry approaches, as well as inorganic chemistry, bacteriology (for FIB) and mycology. The outbreak investigation required cooperative engagement of multi-disciplinary analytical teams.

Regardless of the cause of concern, mitigating actions should take place in the face of beach contamination events. In the case of human faucal pollution, there is always a concern of transmission of residual water-borne pathogens, e.g., enteric viruses, pathogenic bacteria, high concentrations of opportunistic fungi, parasites, and anti-microbial resistance genes (ARGs).

### 5.2. Microbial Source Tracking (MST)

In the case of a surge in FIB, the inevitable question is: “Where did it come from?” Sand contaminated by water and vice versa has been extensively addressed in Whitman et al. [22], who described dispersion, survival, predation on, and ultimately, the fate of the FIB in sand and water. However, in addition, the ultimate origin of the FIB is important to determine. Naturally, FIB that indicate human excreta is a clear warning that human pathogens may be present and cause harm to humans. These are the most relevant FIB for human health, but not the only ones. Other FIB that indicate non-human excreta or different biological groups may indicate possible zoonoses (like bird flu), haemorrhagic *E. coli*, and ARGs. It is thus desirable to be able to track the origin of FIB in sand and in water during a surge, and hopefully mitigate it upstream.

In 2000, Bernhard and Field [90] developed a way to characterize fecal contamination from cows and humans, based on the 16S subunit of the ribosomal DNA, of the genus *Bifidobacterium* and the *Bacteroides*-*Prevotella* group. They reported PCR primers that differentiated both specific biological groups. The aim of their research was mainly to help identify and mitigate diffuse (non-point-source) fecal pollution in water, which tends to be associated mainly with run-off carrying fecal contamination from cattle in wet regions of the most developed parts of the world.

Microbial Source Tracking (MST) tools have since been extensively studied and applied, especially to discriminate the origin of water pollution events. The main objective of this method is the differentiation between human and non-human fecal contamination sources. Due to inter-species barriers to transmission, pathogens of human excreta represent the main public health concern.

Initially, MST was mainly based in library-dependent methods, where a database of fecal samples from known hosts were typed on an isolate-by-isolate basis and compared to the fecal sample under analysis. Currently, library-independent methods using PCR or quantitative PCR (qPCR) to target specific genes of host-associated bacteria are preferred [91]. This latter approach is possible due to the extensive effort in developing primers sets that are specific to several biological groups, including humans [92], seagulls [93], cattle [94], and dogs [95].

There are only a few sporadic reports using this methodology to unravel the contaminations source(s) in sand [42,96], however its potential has widespread support and should be further explored. Valério et al. [16], for instance, assessed a suspected case of fecal contamination of unknown origin that resulted from multiple contributions (horses, seagulls, and dogs). This study is a clear example of how using an MST approach for sand can help elucidate the different contamination sources present at a beach.

Other methods for MST, namely culture independent technologies, are also in use given the advance of real-time qPCR and especially using next generation sequencing (NGS) based DNA analyses, but the financial and computational costs of these methods can be high [97].

### 5.3. Analyzing Microbes in Beach Sand

#### 5.3.1. Fecal Indicator Bacteria (FIB)

The FIB in sands can be analyzed by standard membrane filtration by plating filters on agar media, or by chromogenic substrate. For the chromogenic substrate, Colilert^®^ and Enterolert^®^ from IDEXX Laboratories, Inc. (Westbrook, ME, USA) represent the most common system used. The principle is to extract the FIB by shaking the sand sample immersed in distilled water, followed by processing the extract as though it is a water sample. The analysis returns the most probable number (MPN/100 mL) of coliforms, *E. coli* and enterococci, which must be reversed to the dilution used in the extraction. In terms of details, Sabino et al. [50] used a 1:10 extraction (sand to distilled water) and 30 min circular shake at 50 rotations per minute (RPM). Boehm et al. [89] recommends a faster approach by extracting 10 g of sand with 100 mL of distilled water or phosphate buffered saline (PBS), followed by membrane filtration of the eluent as if it were water [98] to yield colony forming units (CFU) per 100 mL. If source tracking is in order, a detailed protocol for MST analyses has recently been described by Valério et al. [16].

#### 5.3.2. Detection of Other Bacteria

Bacteria, in general, may be difficult to isolate depending on the level of contaminants within an environmental sample. Nonetheless, selective media and specific temperatures can be useful to isolate bacteria of interest. Following the same system as for FIB, Pseudolert^®^ (also manufactured by IDEXX Laboratories, Inc., Westbrook, ME, USA) can be used for the detection and count of *Pseudomonas aeruginosa*. Other bacteria necessitate specific isolation techniques separate from those manufactured by IDEXX. The scientific literature is scarce on bacteria other than FIB. However, in 2009, Goodwin and Pobuda [60] studied the presence of MRSA by performing a novel procedure. Two grams of sand was vigorously hand shaken in 10 mL of PBS before being vacuum filtered through a sterile 30 μm, 47 mm nylon net filter (Millipore, Beford, MA, USA). A 10 mL PBS rinse removed any residual sand from the shaking container. This process was repeated until enough sand-water solution was generated for membrane filtration. The sand–water solution was also homogenized via hand mixing prior to filtration. After filtration, the filters were incubated on SCA or C-MRSA selective and differential media for *S. aureus* and MRSA, respectively (BD Biosciences, San Jose, CA, USA) [46]. Recently, a review on ESKAPE pathogens (*Enterococcus faecium*, *Staphylococcus aureus*, *Klebsiella pneumoniae*, *Acinetobacter baumannii*, *Pseudomonas aeruginosa*, and *Enterobacter* spp.) in environmental reservoirs, such as surface water, wastewater, food, and soil, addressed these isolations from beach sand [99]. Particularly, Akanbi et al. isolated MRSA from beach sand in the Eastern Cape Province of South Africa in 2015–2016, followed by isolation and molecular identification, and testing resistances via the Kirby-Bauer disk diffusion method on Mueller-Hinton agar [63].

#### 5.3.3. Isolation of Fungi

Sabino et al. [50] describes analytical methods for fungi, whereby sand is extracted with water in a 1:1 ratio, in low energy, shaking orbitally at 100 rpm for 30 min. Sample triplicates are then plated. Malt yeast agar with chloramphenicol is used for all species and Mycosel agar (with chloramphenicol and cycloheximide), specifically for dermatophytes. Sample plates are then incubated for 5 and 15 days, respectively, at 27.5 °C. After incubation, plates undergo tentative colony identification via picking and counting one colony of each morphotype, and dividing them into yeast-like species, opportunistic and allergenic species, and dermatophytes as three parameters to assess fungi. The result is the average count of the triplicates for each parameter, per gram of sand. This method is rather laborious and time consuming but provides a complete analysis of the culturable mycobiota in a sand sample.

Unlike yeasts that form mainly budding cells, molds usually occur as hyphae. If broken, any hypha will start a new colony. It is thus truly relevant how to approach fungal analyses of sand, considering that there will always be a trade-off between aqueous extraction from sand and breakage of the hyphae during the process due to vigorous agitation. In light of this trade-off, Sabino et al. [50] opted for orbital shaking with a speed of 100 rpm. The same laboratory is still doing so today, followed by plating in malt yeast agar, supplemented with chloramphenicol (0.05%), for all fungi and Mycosel agar, supplemented with chloramphenicol and cycloheximide. The latter is used specifically for dermatophytes, given the growth speed reduction of fast-growing fungi, which allows dermatophytes to grow and be visible, instead of being overtaken by fast growing molds.

In mycology, it is necessary to use a medium to recover as much of the fungi of interest as possible. Sabouraud with chloramphenicol is the traditional medium of choice as it is not selective. Yet, samples with an abundant presence of *Mucorales* may require the use of a medium with Dichloran Glycerol Agar combined with Rose Bengal to inhibit their excessive growth [100]. This is frequently the case with inland beaches due to the stronger presence of vegetable matter since many species of this order are plant pathogens [101]. Coastal beach sands may also yield some isolates of *Mucorales* but usually only when highly contaminated with fungi from many species.

Incubation needs to extend long enough to allow as many fungal colonies as possible to become visible, but not too long to have the faster growing ones cover the slower growing ones. Again, there are trade-offs for isolating mixed cultures of unknown species from environmental samples: many dematiaceous fungi are slow growing, including dermatophytes and *Exophiala* spp. [102], which cause phaeohyphomycosis. Conversely, *Mucorales* and *Trichoderma* spp. are extremely fast growers [100,103]. Moreover, yeasts grow by cell division on a growth medium, not by releasing hyphae. Hence, molds spread onto the culture media while yeast colonies have a more restricted growth. In addition, there is competition between both groups. For example, the statins [104], used for pharmaceutical purposes in humans to lower cholesterol levels in blood, are in fact metabolites produced by molds, intended to slow down the growth of yeasts and even of other species of molds. The statins interfere with the production of ergosterol, necessary for their growth [26]. The inoculum from an environmental sample should thus be diluted enough to allow all species to grow without confluence, during a long enough incubation period, and at a temperature that is appropriate for the intended purpose. However, dilution of the inoculum to ensure a robust analytical disposition of all colonies will inevitably lead to the loss of species present at lower levels [105].

Sand analysis for health protection should target a temperature that matches that of the surface of the human body. The approach used in bacteriology of selecting the pathogens from the bacteria that can grow at 37 °C does not apply to Mycology. Keratinophilic fungi infect keratinized tissue, including skin, nails, and hair. A temperature of 27.5 °C will allow all medically relevant fungi to grow [50].

Fungi in water were addressed in Brandão et al. [65], but the conclusion of the study was that more data were needed before recommendations could be issued. The same team is preparing to address this data gap in the future.

#### 5.3.4. Identification and Taxonomic Classification of Fungi

Fungal taxonomy is currently undergoing a revolution, mainly due to new data arising from the use of molecular biology techniques [106]. Still, the primary approach tends to remain identification by micro and macro characteristics of the colony. Some of the common fungi found in the beach environment are readily recognizable at first glance. Microscopic verification of the typical structures may help to distinguish, for example, some *Penicillium* and *Aspergillus* section *Fumigati* species. Yeasts, however, require additional biochemical testing to be differentiated. Sabino et al. [50] described in detail how sand analysis can be performed for the identification of the fungi and bacteria. Although the same institution currently identifies many of the fungi by matrix-assisted laser desorption/ionization-time of flight (MALDI-ToF) or molecular approaches based on sequencing of the ITS1 and ITS2 regions of the ribosomal DNA. MALDI-ToF is fast and requires little handling before an acceptable identification is achieved. Molecular identification tends to be broader in the number of species that can be identified, but it is more costly and labor intensive. The process requires DNA extraction, purification, and amplification with the primers chosen for the identification intended by PCR, cleaning of the PCR product of the amplification, followed by sequencing of this product and searching for and aligning (blasting) the sequences with DNA databases [65]. This system allows the clear distinction of many cryptic species, depending on the primers chosen to help amplify the target DNA, as shown in the study of Novak Babič et al. [107].

### 5.4. Analyzing Insects and Helminths in Beach Sand

There is currently no methodology published for detecting insects in beach sand. However, WHO [11] lists the main groups considered of interest, which are all Diptera: mosquitoes, biting midges, sand-flies, flies, and blackflies. When flies and blackflies occur in large numbers, they are a nuisance and mosquito, midge, and sand-fly bites can be painful in addition to being vectors of disease for humans and animals.

Mosquito numbers typically increase during wet weather or following tide triggers and they are often most active at dawn. It is recommended that beachgoers try to avoid exposure outdoors at sunset and overnight; they should wear long sleeves and use insect repellent. As mosquitoes breed in standing water, getting rid of devices that hold water is a simple way to stop them from breeding [108].

Biting midges are a common nuisance along certain coastal areas and are most active in intertidal zones, including canals, rivers, and estuaries. Sand-flies tend to be found on beaches where the sand is slightly earthier and are usually a problem when people lie directly on the sand for long periods of time. To help reduce nesting sites, it is advisable for authorities to remove algae and other debris brought to sands by storms. As biting midges and sand-flies are sub-optimal fliers, effective protection can be provided through the use of fans in public places to prevent bites, combined with repellents and loose clothing [109].

Nuisance flies are attracted to dead animals, feces, and garbage, allowing them to spread a variety of disease-causing bacteria and parasites. Some flies can deliver painful bites and transmit diseases to humans and animals. Good sanitation, by eliminating flies’ breeding sites, is the main fly control recommendation at beach facilities. Thus, garbage containers must always be closed and fecal matter from dogs and other animals should be eliminated. When these dipterans become pests, public health entities and municipalities can implement chemical control programs with insecticides, such as pyrethrins and pyrethroids [110], while preventing the development of potential insect resistance by alternating the products.

Helminths are currently monitored only in Lithuania, where the beach sand is tested once before the start of the bathing season and at least four times during the bathing season. The procedure is described in detail by the Ministry of Health of the Republic of Lithuania [41]. Ideally, this regulation should be tested and implemented in other countries as part of beach sand monitoring activities with relevance for human health protection.

## 6. Beach Sand Quality—Current Situation

One of the objectives of this review was to describe strategies to assess beach sand quality. Historically, beach sand had not been addressed although it has been long recognized since the 1960s as a fomite for human infections and a source of FIB in bathing waters. Beach sand quality did not become a regulatory concern until 2010, with the Portuguese Blue Flag program, in collaboration with the National Institute of Health Doctor Ricardo Jorge and the Portuguese Environment Agency. Between 2006 and 2011, this cooperation led to the monitoring of beach sand across Portugal [50]. The effort found that sampling before the bathing season yields higher counts of microorganisms and yeasts, and dermatophytes, which indicate intensive human use. There is a microbial cumulative effect of beach use throughout the bathing season and there is a significant correlation between *E. coli* and *C. albicans* and between enterococci and *C. albicans*. The final finding confirmed the human nature of the contamination since *C. albicans* is nearly human specific. Since the end of the implementation in 2016, the results may have changed significantly due to the investment made by all the member-states to provide excellent water quality at bathing waters by improving the treatment of residual waters. Since this program came to an end, no other known monitoring program has taken place regarding sand. Based on the WHO recommendations of 2021 [11], the Blue Flag program of Portugal implemented sand monitoring as new awarding criteria for the 2022 bathing season. Knowledge in this field, however, has improved immensely with publications arising from all over the world. It is thus time to develop regulations to help guide and set a common worldwide plan of action to integrate sand quality in beach management tools.

Educating the public and beach managers on how to maintain a healthy level of potential pathogens, is a crucial element in maintaining the good health of the beach and beachgoers. WHO [11] describes a certain number of actions in this perspective (Table 1).

These measures alone may not maintain sand microbiota at acceptable levels all the time, but they are informed by years of studies on diffuse pollution and hygiene concepts targeting human health protection in recreational water environments.

## 7. The Way Forward

In addition to research to fill the many gaps that currently exist and are discussed throughout this text, it is necessary to develop a procedure to classify beaches according to their results of sand monitoring, as currently happens with water. A future health-oriented set of safety parameters for fungi depends on epidemiological studies or quantitative microbial risk assessment (QMRA) estimates. Considering the non-normal distribution of fungal counts over time, using standard deviations and geometric means is not advised. A good alternative is to classify a beach as compliant or not compliant, allowing a certain number of results to fail, in the case of ordinary microbiota fluctuation. A 20% rejection rate, for example, is not unreasonable in the case of fungi, according to Brandão et al. [65]. This would lead to a guidance value of 89 CFU/g of total fungi in sand and a rejection limit of the 80th percentile, which is 490 CFU/g. This means that during a sampling period, values above 490 CFU/g are acceptable in ≤20% of the samples.

For enterococci, the value stated in WHO [11] theoretically reflects the same health effect of consuming water during beach bathing. Therefore, care should be taken if samples exceed 60 CFU/g of sand. This value is considered provisional as it is the result of a QMRA calculation which does not consider the native flora of a beach. Epidemiological studies should be conducted to confirm the validity of assumptions of the calculation. In large freshwater basins, FIB may not reflect human fecal contamination, and the source can be further assessed by studying the microbiota at a genetic level, including using MST testing. With an understanding of potential sources of microbes, beach management strategies should be developed using this information to lower microbial concentrations in sands to levels below the guidelines.

In the future, epidemiological studies combined with sand microbiological analysis and additional relevant parameters should be prioritized. The values used for the new WHO guidelines [11] are currently provisional due the lack of clinical confirmation. One option would be to conduct a self-assessment study of a large population visiting defined beaches, similar to Leonard et al. [111]. Studies should also focus on exposures to endemic fungi, bacteria other than MRSA and *P. aeruginosa*, and other pathogens of emerging concern in beach recreational water settings. Such studies should integrate culture-based and molecular quantification of relevant indicators of sand quality.

## Figures and Tables

**Figure 1 ijerph-20-05710-f001:**
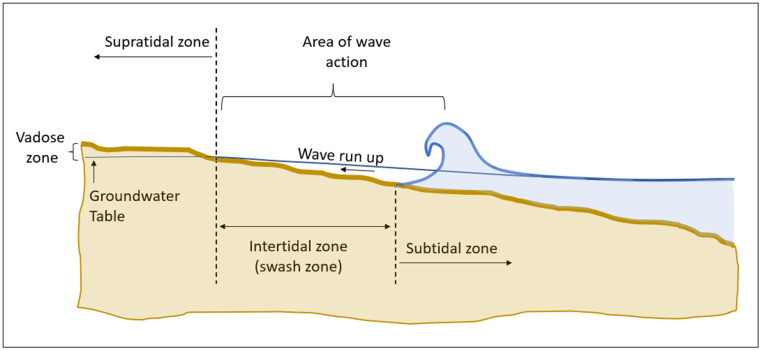
Definition of beach sand zones.

**Table 1 ijerph-20-05710-t001:** Possible actions to improve sand quality.

When the sand quality is documented as poor and action needs to be taken, chlorination is the most used approach. Biocides, if needed, do not go beyond the superficial sand layers, unless a dispersing agent is added [22]. If too polluted, sand may be replaced as a remediation approach [29].
Implementation of covered garbage receptacles that are animal-proof and protected from precipitation, in designated areas.
Proper solid waste management to prevent non-native bird usage at the beach.
Toilet facilities that can minimise contamination impacts from human faecal matter and encourage proper personal hygiene at the beach.
Appropriate design and implementation of drainage systems for beach areas so that drainage from parking lots and backshore areas cannot flow into the nearshore beach zone [30].
Re-direction of stormwater drainage from nearby communities away from the beach, combined with limited recreational access to downstream areas if stormwater does reach the beach area, to minimise exposure to beachgoers.

**Table 2 ijerph-20-05710-t002:** Pathogens, opportunistic microorganisms and faecal indicators reported from beach sand.

Cases Reported	Details
*Staphylococcus aureus* and MRSA Increased persistence in California, USA [48]	Found in 2.7% of the samples
*Candida* spp. including *C. auris.* increased presence with decreased precipitation (predominantly found in dry sands) in Portugal, Florida, USA and Colombia [49,50,51]	“85 of 495 samples were positive for *Candida* spp.” [49], “*C. auris* isolated from water and harsh wetlands” [50]; “Association between indicators and pathogens; *C. tropicalis* was the most frequent *Candida* spp. isolated from dry sand with an average count of 34.3 CFU/g of dry sand, followed by *C. parapsilosis* with 1.6 CFU/g, *C. glabrata* with 1.2 CFU/g and *C. guilliermondii* with 0.6 CFU/g” [51].
*E. coli* elevated numbers in water correlated with increased wave height in Lake Huron, Canada [52] and associated with wider dispersal of microbes and water releases into Lake Superior, USA [53]	*“E. coli* concentrations in the surface (unsaturated) sand 1 m landward of the initial shoreline (P1) were 1.23 ± 0.99 log CFU/g” [52]; “When *E. coli* concentrations in sand and sediment samples were converted to CFU per interstitial water, the greatest numbers of *E. coli* were observed in nearshore and upshore sands, followed by shoreline sands and sediment. These numbers were, on average, 63, 74, 1087, and 4982 times greater in sediment, shoreline, nearshore, and upshore sand samples, respectively, than the concentration of *E. coli* in lake water expressed as CFU/mL” [53].
Enterococci in mobilized sand caused a spike in water contamination with increasing wave action in Florida, USA (experimental, laboratory conditions) [54,55]	“The time duration for a certain percent die-off is inversely proportional to the solar radiation intensity. It takes 2.2 h to deactivate 90% of the total enterococci at noon time with a near maximum solar insolation of 800 W m^−2^“ [54]; “For trials with waves, analysis of the top and bottom layers of sediment in the ‘‘final’’ seeded sand revealed that more enterococci were removed from the top layer (78% removal and standard deviation of 17%) as compared to the bottom layer (58% removal and standard deviation of 28%) (*p* = 0.05, average difference between top and bottom layers = 194 CFU/g dry sand)” [55].
Helminths found in Lithuanian beach sand [41]	“The soil must not be contaminated with helminths, enteroviruses, pathogenic enterobacteria, intestinal rods.”
Hepatitis A virus (HAV) and Human Adenovirus (HadV) genetic material found in beach sand with qPCR, although viability of viral particles was not tested [56]	“HAV were detected in 6.25% (3 out of 48) and HadV were detected in 8.33% (4 out of 48) samples using qPCR. These results corroborate the reporting of sand as an independent source of GI illness.”

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
