# Peer review of "Strategies for Monitoring Microbial Life in Beach Sand for Protection of Public Health"

_ijerph, 2023, doi:10.3390/ijerph20095710_

Round 1

Reviewer 1 Report

It is an interesting review on a subject that deserves more attention. However, some issues need to be addressed and improved before publication.

Spelling and grammar should be reviewed.

The title should reflect that the review has an emphasis on microbiology. 

Please avoid using the term 'microbial flora' 

A table providing the most relevant microbiological data should be included.

Author Response

We appreciate the reviewer’s comments and have made edits to the manuscript (see tracked changes version) in response to the reviewer’s comments. We appreciate these comments and believe that it has improved the quality of the paper.

Response to Comments from Reviewer 1

Q: It is an interesting review on a subject that deserves more attention. However, some issues need to be addressed and improved before publication.

R: Thank you, on behalf of all the authors.  Below we provide point-by-point responses to each issue listed.

Q: Spelling and grammar should be reviewed.

R: The spelling and grammar has been checked throughout.  Edits are showed tracked.  Examples are shown at lines 21, 25, 27, 28, and so on.

Q: The title should reflect that the review has an emphasis on microbiology. 

R: Agreed. The New title is “Beach sand microbiology monitoring strategies for protection of public health”.  The word “microbiology” has since been added to the title.

Q: Please avoid using the term 'microbial flora' 

R: The term has been removed in all three sites and replaced with either microbial community (lines 118 and 137) or microorganisms (line 562). 

Q: A table providing the most relevant microbiological data should be included.

We have since converted the information in section 3 into tabular format.  The caption for the new table is, “ Possible actions to improve sand quality.” (line 187).

Reviewer 2 Report

- Remove it from lines 31 to 56 

- Are the conclusions supported by the evidence and arguments presented, and do they address the main question?

- In line 72, substitute bacteria, fungi, parasites, and viruses.

- Any scientific nane should be italicized.

- in line 82, change this paper to a previous study by Fujioka..... and include the number of references 

- In line 609, 0.05%/L is incorrect. Change it to 0.05% only.

- In line 620, mucorales is order, and then no italic is written.

- You can combine fungi and isolation and identification of fungi, faceacal bacteria,  other bacteria and isolation and identification of bacteria under one subtitle for type of microorganisms. no repeat

Author Response

Q:- Remove it from lines 31 to 56 

R:  We have since retitled the outline to be more descriptive.  The title was changed from “CURRENT OUTLINE BELOW FOR REFERENCE” to “Contents”.  With this new title we believe it is clearer that the list of sections is to help guide the reader through the contents of the paper. (line 32)

Q: Are the conclusions supported by the evidence and arguments presented, and do they address the main question?

R: We have since adjusted the text such that the goals of the manuscript as described in the abstract match the conclusions.  The edits include removing reference to “public health management” (line 24) in the abstract as this section has since been removed.  We tie the concluding section called “the proposed way forward” more tightly with the abstract by stating it again at line 25 in the abstract.  The second to the last section (section 6) has been more closely aligned with the objectives stated in the abstract by emphasizing that, “One of the objectives of this review was to describe strategies to assess beach sand quality” (lines 821 to 822).

Q: In line 72, substitute bacteria, fungi, parasites, and viruses.

R: Removed. It now reads simply “Between 1969 and 2003, many publications addressed sand in its

multiple microbial fronts” (lines 76 to 78).

Q: - Any scientific nane should be italicized.

R: We have since gone through the entire document and have italicised all scientific names.

Q: - in line 82, change this paper to a previous study by Fujioka..... and include the number of references 

R: Two additional references have been added citing Dr. Fujioka’s earlier research focused on alternative indicators. They include Fujioka et al. 1997 (Appropriate recreational water quality standards for Hawaii and other tropical regions based on concentrations of Clostridium perfringens) and Fujioka and Zhizumura 1985 (Clostridium perfringens, a reliable indicator of stream water quality). (line 87 references 5, and 6).

Q: - In line 609, 0.05%/L is incorrect. Change it to 0.05% only.

R: Done. Thank you for noticing this. (line 721).

Q: - In line 620, mucorales is order, and then no italic is written.

R: All scientific names have been checked to be italicised. (lines 729, 733, 740 as examples)

Q: - You can combine fungi and isolation and identification of fungi, faceacal bacteria,  other bacteria and isolation and identification of bacteria under one subtitle for type of microorganisms. no repeat

R: Subtitle 5.3 has been generalized to now read as “Analysing microbial presence in beach sand”.  The word “bacteria” was replaced by “microbial”.  By doing this we were able to remove the second subtitle 5.4 which focused on fungi, thereby removing some of the repetition.

Reviewer 3 Report

The review entitled " Beach sand monitoring strategies for protection of public health " written by João Brandão, Elisabete Valério, Chelsea Weiskerger , Cristina Veríssimo , Konstantina Sarioglou , Monika  Novak Babič, Helena Solo-Gabriele, Raquel Sabino, and Maria Teresa Rebelo, concerns the summary of biological indices and strategies to assess beach sand quality and monitoring, and also reports the proposed way forward for sand biological monitoring approach.

The meaning of the article is comprehensible resulting really interesting and the paper gives a precious contribute to the knowledge on this field.  Anyway, I have minor revisions to recommend:

1.       Since the title and the abstract have to be representative of the study you perform, I suggest to specify in both that the monitoring you are referring to is only the biological one.

2.       Please improve quality of figure 1 since writings are showed in a weird format.

3.       Paragraph 3. The purpose of this first paragraph is not clear. Please add a subheading that explicits it´s own meaning. Moreover, a better introduction of remediation/bioremediation methods is needed. Indeed, the section from line 165 to line 179 appears too brief and confusing. You mentioned chlorination and biocides and then jumped on biofilms in laboratory.

4.       Line 235. It seems that the sentence is not finished. Please add some “namely” examples.

5.       Paragraph 3.2. Please add examples of detection limit values stated by legislation for enterococci and E. coli in water.

6.       Line 286-288. Antibiotic resistance bacteria and pathogens are increasingly found in beach and marine environment representing a troublesome issue for human health that should not be underestimated. I suggest to add few more lines on these taxa that for sure have to be monitored.

7.       Line 395-400. Please rewrite it more clearly, since it is confusing.

8.       Paragraph 5.1. The title is incomplete and not clear or representative of the content. What are you referring to? Moreover, did you describe these recommended analyses in the “routine” paragraph? please specify it as you have done in other paragraphs.

9.       Paragraph 5.1.1. Also in this casw, the title is not clear. Please rewrite it to be more explicit. Are you describing analysis that are routinely carried out or that the authors recommended to perform as routine analyses?

10.   Line 450-451. It seems some words are missing. Please check the grammar of the sentence.

11.   Line 487-492. I recommend to move these lines to the paragraph where you describe in detail MST, and only introduce it here explaining that you are going to describe and deepen it later, avoiding to be redundant.

12.   Paragraph 5.3. I suggest to move this paragraph to the beginning of the section 5 since it deals with the primary step: sample preparation, management, handling and quality, before it undergoes microbial analyses.

13.   Paragraphs 5.4-5.5-5.6. I recommend to merge this section to the paragraph above (5.1.1) to not be repetitive. You already introduced analyses for microbial detection and described some of them (see lines 426-443). Indeed, some information are reported twice.

14.   Lines 732-742. Please add full stops or semicolons at the end of each sentence.

15.   Please when possible, cite more up to date references.

Author Response

Q: The review entitled " Beach sand monitoring strategies for protection of public health " written by João Brandão, Elisabete Valério, Chelsea Weiskerger , Cristina Veríssimo , Konstantina Sarioglou , Monika  Novak Babič, Helena Solo-Gabriele, Raquel Sabino, and Maria Teresa Rebelo, concerns the summary of biological indices and strategies to assess beach sand quality and monitoring, and also reports the proposed way forward for sand biological monitoring approach.The meaning of the article is comprehensible resulting really interesting and the paper gives a precious contribute to the knowledge on this field.  Anyway, I have minor revisions to recommend:

R: The authors thank the reviewer for this opinion

Q: 1.       Since the title and the abstract have to be representative of the study you perform, I suggest to specify in both that the monitoring you are referring to is only the biological one.

R: pertinent point, also pointed out by another reviewer so we adjusted the name of the paper to: “Beach sand microbiology monitoring strategies for protection of public health” and the abstract lists bacteria and fungi as monitoring parameters.

Q: 2.       Please improve quality of figure 1 since writings are showed in a weird format.

R: The figure has since been adjusted. Color has been added to more easily distinguish sand versus water.  The line types and thicknesses have been adjusted to better distinguish lines associated with labeling in comparison to lines that are part of the image.

Q: 3.       Paragraph 3. The purpose of this first paragraph is not clear. Please add a subheading that explicits it´s own meaning. Moreover, a better introduction of remediation/bioremediation methods is needed. Indeed, the section from line 165 to line 179 appears too brief and confusing. You mentioned chlorination and biocides and then jumped on biofilms in laboratory.

R: Agreed. We removed the following text “When the sand quality is documented as poor and action needs to be taken, chlorination is the most commonly used approach. Biocides, if needed, do not go beyond the superficial sand layers, unless a dispersing agent is added [20]. If too polluted, sand may be replaced as a remediation approach [25].” This information has since been moved to a table (Table 1) in order not facilitate the logical flow of the paper. (line 187)

Q: 4.       Line 235. It seems that the sentence is not finished. Please add some “namely” examples.

R: This line has now been merged with the previous sentence to clarify that the only existing regulation is the one listed in the paragraph. (line 261).

Q: 5.       Paragraph 3.2. Please add examples of detection limit values stated by legislation for enterococci and E. coli in water.

R: Considering that the WHO recommends only enterococci, not E. coli, The authors found appropriate to add the following sentence in that section: “It also is currently in use in regulation, e.g. the European Bathing Water Directive [8].” (line 303).

Q: 6.       Line 286-288. Antibiotic resistance bacteria and pathogens are increasingly found in beach and marine environment representing a troublesome issue for human health that should not be underestimated. I suggest to add few more lines on these taxa that for sure have to be monitored.

R: We have since added a sentence and references addressing antifungal resistant Candida auris and multi-resistant yeast. (lines 323-325).

Q: 7.       Line 395-400. Please rewrite it more clearly, since it is confusing.

R: It has been reviewed and now reads, “Should incremental sampling not be possible, a composite can be collected for beaches less than a few hundred metes in length.  The composite sample would consist of three grab samples spanning the length of the beach.  These three grab samples can then be combined, homogenised, and analyzed as one sample [66].  If beaches are more than a few hundred meter in length, in which case, the beach should first be divided into areas and each area considered a separate beach.” (lines 447 to 454).

Q: 8.       Paragraph 5.1. The title is incomplete and not clear or representative of the content. What are you referring to? Moreover, did you describe these recommended analyses in the “routine” paragraph? please specify it as you have done in other paragraphs.

R: Thank you very much for noticing. Fully agreed. The new title is now “Two tiers analytical approach.” (line 472)

  1. Paragraph 5.1.1. Also in this casw, the title is not clear. Please rewrite it to be more explicit. Are you describing analysis that are routinely carried out or that the authors recommended to perform as routine analyses?

R: Also, fully agreed. We have since reorganized the information presented in this section. The new title is now “Analysis during outbreak conditions.” (line 482)

Q: 10.   Line 450-451. It seems some words are missing. Please check the grammar of the sentence.

R: The text has been reworded.  It now reads as, “Upon on-site description of the problem that originated the study, the local team described how a tropical winter storm delivered enormous amounts of near coast debris into the cove, including high quantities of decaying vegetable matter. The debris was reflected in the fungal analysis, by the considerable presence of Fusarium spp, of Aspergillus section Circundati, and some Candida tropicalis which are common plant pathogens and colonisers.” (lines 505 to 510).

Q: 11.   Line 487-492. I recommend to move these lines to the paragraph where you describe in detail MST, and only introduce it here explaining that you are going to describe and deepen it later, avoiding to be redundant.

R: Recommendation accepted. Those lines were moved to the MST section where it now reads: “There are only a few sporadic reports using this methodology to unravel the contaminations source(s) in sand [37, 78], however its potential has widespread support and should be further explored. Valério et al. [14], for instance, assessed a suspected case of faecal contamination of unknown origin that resulted from multiple contributions (horses, seagulls and dogs). This study is a clear example of how using an MST approach for sand can help elucidate the different contamination sources present at a beach.” (lines 631 to 637).

Q: 12.   Paragraph 5.3. I suggest to move this paragraph to the beginning of the section 5 since it deals with the primary step: sample preparation, management, handling and quality, before it undergoes microbial analyses.

R: Agreed. It has been moved to after routine analysis and the new sections now has the following organisation “5.1.1. Routine analysis of sand, 5.1.2. Quality Assessment Schemes, 5.1.3. Outbreak and full population analysis.” (lines 482, 539, 563)

Q: 13.   Paragraphs 5.4-5.5-5.6. I recommend to merge this section to the paragraph above (5.1.1) to not be repetitive. You already introduced analyses for microbial detection and described some of them (see lines 426-443). Indeed, some information are reported twice.

R: We have since merged the information. We have merged the analytical details described in the first paragraph under section 5.1.1. (lines 573 to 492) with the section (now numbered 5.3.3) which focuses on fungal analytical details and with a second section (now numbered 5.3.1) which fcouses on FIB analytical details.  We also removed text at lines 503 to 518 which describe the extraction method for microbes from sand as it was explained in the FIB section (now numbered 5.3.1).  The deletions of text can be observed at lines 473 to 492 and at lines 512 to 527. The new integrated text is at lines 666 to 672 and at lines 694 to 704. Thank you for this comment as it helped us to reduce the repetition between these sections.

Q: 14.   Lines 732-742. Please add full stops or semicolons at the end of each sentence.

R:  This text has since been deleted and the reader is redirected to the content in a new table (Table 1). (line 187).

Q: 15.   Please when possible, cite more up to date references.

R:  9 new references have been added.  They include Laszlo 2022, Escandón 2022, Abdool-Ghany et al. 2022, Abdool-Ghany et al. 2023, Sahwell et al. 2021, Steffen et al. 2023, Akanbi et al. 2017, Cogliati et al. 2022, and Denissen et al. 2022.

Round 2

Reviewer 1 Report

No further comments 

Author Response

Thank you for helping us improve our manuscript

Reviewer 2 Report

in line 621, Mucorales this is order not genus, and then it should not be italic

Author Response

Thank you for noticing once again. All 4 Mucorales throughout the manuscript have now been deitalicised.